# Use of the Ultrasound Technique as Compared to the Standard Technique for the Improvement of Venous Cannulation in Patients with Difficult Access

**DOI:** 10.3390/healthcare10020261

**Published:** 2022-01-29

**Authors:** Ángeles Rodríguez-Herrera, Álvaro Solaz-García, Enrique Mollá-Olmos, Dolores Ferrer-Puchol, Francisca Esteve-Claramunt, Silvia Trujillo-Barberá, Pedro García-Bermejo, Jorge Casaña-Mohedo

**Affiliations:** 1Department of Health Sciences, Faculty of Health Sciences, Universidad Europea de Valencia, 46010 Valencia, Spain; mariaangeles.rodriguez@universidadeuropea.es (Á.R.-H.); alvarojose.solaz@universidadeuropea.es (Á.S.-G.); silvia.trujillo@universidadeuropea.es (S.T.-B.); pedro.garcia@universidadeuropea.es (P.G.-B.); 2Neonatal Research Unit, Health Research Institute La Fe, 46026 Valencia, Spain; 3Division of Neonatology, University and Polytechnic Hospital La Fe, 46026 Valencia, Spain; 4Emergency Department, Hospital de La Ribera, 46600 Valencia, Spain; enrique.molla@uv.es (E.M.-O.); lolesferrerpuchol@gmail.com (D.F.-P.); 5Department of Health Sciences, Universidad Internacional de Valencia, 46002 Valencia, Spain; jorge.casana@ucv.es

**Keywords:** emergencies, nursing peripheral cannulation, ultrasound

## Abstract

(1) Objective. We aimed to demonstrate that the use of the ultrasound-guided technique facilitates peripheral venous cannulation as compared to the standard technique in patients with difficult access at emergency services. (2) Method. A case–control study, randomized research. Variables were collected from a population with non-palpable or not visible veins, classified into size risk groups for 6 months. In the comparative analysis, the patients were divided into two groups: the cases group was composed of patients to whom the peripheral venous cannulation was performed with the ultrasound-guided technique (UST), while the control was composed of patients with whom the standard technique (ST) was performed. The ultrasound LOGIQ P5 750VA from General Electric Healthcare, with an 11 mHz linear probe, was utilized, along with peripheral venous catheters model Insyte^TM^ Autoguard^TM^ with gauges of 14G to 26G. (3) Results. Seventy-two cases. The use of the ultrasound decreased the time (618.34s ST, 126s UST) and the number of punctures (2.92 ST, 1.23 UST); about 25% of the patients did not have complications with the UST, as compared to 8% with the ST. The use of the ultrasound decreased the pain experienced by 1.44 points in the visual analog scale, as compared to 0.11 points with the ST. The rate of success of the first try with the UST was 76%, as compared to 16% of the ST. The gauge of the catheter increased with the UST, with successful cannulations obtained with 20G (56%) and 18G (41%) gauges. (4) Conclusions. The use of ultrasound facilitates venous cannulation according to the variables of the study. The ultrasound visualization of the vessels is associated with the selection of the catheter gauge. There was no relation between the complications and the depth of the blood vessels.

## 1. Introduction

Peripheral venous cannulation is a very common technique utilized for diagnosis and therapeutic aims, one that is considered a fast, safe, and easy-to-perform technique. Among its advantages, we find the simplicity of its placement and only a small number of associated complications, with an estimated number of 90% of patients with a peripheral venous device at Emergency Services (ES). However, with patients with difficult venous access, other more sophisticated techniques or alternative manners of performing this technique are needed to ensure good quality care [1].

A large number of patients are usually treated at ES, of which a large percentage will have a difficult venous access. Nevertheless, the vein anatomy does not always follow the same norms, and caregivers must often deal with patients who will present difficulties in the cannulation of their veins, due to their veins being sinuous, narrow, indurated, or fragile and non-palpable, or non-visible. Other factors must be considered, such as the depth of the vessels or the prior pathologies of the patient, which will have an influence on their vascular capital, or even vasoconstriction due to the cold, hypotension, or shock, as well as in any other technique, the knowledge and experience of the person performing it [2,3]. 

These patients are defined as difficult-to-access or difficult venous access patients and are a placed in risk groups according to different studies and according to the parameters established in the A-DIVA scale [2,4,5,6,7].

To ease cannulation, one can utilize different methods or applications and light-based devices, or the use of ultrasound (US) can also be used to favor the cannulation of difficult venous access patients. With these patients, in case of a failed peripheral venous access, other types of approaches are used, such as external jugular cannulation or central venous catheters (CVC), having in mind the frequent risks associated to them, either linked with the puncture, such as arrhythmias, artery perforation, or hematoma [8], or others such as pneumothorax, fracture of the line, misplacement, migration, infection, or thrombosis [9]. International studies have provided evidence that with a brief but complete period of training, ES technicians can successfully perform an ultrasound-guided peripheral intravenous cannulation of patients with a difficult venous access, avoiding central accesses [10]. Moreover, diverse studies on the insertion of a peripheral venous cannula guided by US, as compared to that based on reference points, showed an improvement in the success of placement through the ultrasound-guided technique, with significant effects on patients with difficult venous access [11].

The appearance of portable ultrasound devices, and even transductors that can connect to mobile phone applications, and the great simplification of their use, have been key pieces in the implementation and development of this tool in ES and mobile medical care units, which lead to the study of peripheral venous cannulation.

The safety and quality of care are established as the basis of health care, as well as the safety of the health professionals. Their effect on the correct measures of hygiene, the constant updating of protocols, the appearance of novel techniques, and the continuous training of the staff are the necessary pillars for guaranteeing the optimum quality of the processes. The use of US for nursing techniques is as an emerging measure for the improvement of care, either for the evaluation of the network of vessels in the cannulation of peripheral devices or central ones of peripheral access, insertion of arterial catheters, measurement of the volume of urine before the placement of a urinary catheter, or the verification of the correct placement of a urinary catheter. 

At present, there are no recommendation guides about the routine use of US in peripheral venous cannulation, as no significant differences have been found when used with patients who have an easy access [11,12]. Nevertheless, the use of US has been proposed as an alternative technique for patients with difficult venous access [13,14]. With this idea in mind, the aim of the present randomized research study was to evaluate if the use of the ultrasound technique (UST) facilitates peripheral venous cannulation, as compared to the standard technique (ST), in patients with difficult venous access, by studying different risk groups and discovering if significant differences exist with respect to the study variables.

## 2. Materials and Methods

### 2.1. Study Design and Setting

A case–control study and cross-sectional study were conducted. A non-probabilistic or incidental randomization was used, on the basis of the availability of the ultrasound machine. The patients were divided into two groups: the cases group was composed of the patients with whom the ultrasound peripheral cannulation was performed, and the control group was composed of the patients with whom the standard technique was utilized. The study was conducted at the emergency services (ES) unit of a level III hospital, between 1st January and 1st June 2017. The target population was composed by patients who visited the ES and who had a difficult venous access. Patients with difficult venous access were defined as those whose veins were not visible/palpable to the nurse.

### 2.2. Study Variables

In first place, the sociodemographic and clinical variables of the patients were collected to determine if the samples were comparable. The data collected were date and time, age, sex, vital signs that could be associated with vasoconstriction (blood pressure and temperature), data about the triage system utilized at the center (tag, discriminator, and priority of care), and diagnosis of the patient when released (through the CIE-10 international code).

The behavior of the variables was compared between the control and cases groups as independent variables, by studying the following dependent variables: number of previous punctures before the successful venous cannulation; rate of success; complications that are directly linked to the puncture; time invested measured in seconds, from the starting of the compressor to the blood backflow to the chamber of the venous catheter, and verification of success; the intensity of pain of the patient with the use of the visual analog scale (VAS) with respect to previous experiences; catheter gauge; and area of venipuncture. Moreover, we evaluated whether the UST had advantages in some of the defined, specific risk groups who had non-palpable or non-visible veins, and who were divided into the following: obese patients, considering a body mass index greater or equal to 30 (BMI > 30); diabetic patients, with an active or ceased treatment with cytostatic drugs; those under treatment with anticoagulants/antiaggregant drugs; users of illegal intravenous drugs; and those with some problems that could imply the inability of use of a limb, such as skin lesions, limb affected by hemiparesis due to a cerebrovascular accident (CVA, stroke), ganglion extirpation of mastectomized women, or having a DAVF (arteriovenous fistula).

Moreover, in the cases group, a study was conducted in which the following descriptive objectives were analyzed: to evaluate the existence of a relationship between the depth of the vessel and the complications, and to determine if the gauge of the catheter could be determined according to the diameter of the vessel cannulated. To avoid biases, only a transversal plane or short axis was utilized (Figure 1). This plane was selected as opposed to the longitudinal one in order to obtain references of the structures adjacent to the vessel selected, as well as to be able to correct the trajectory of the needle through the use of the “ring down” method. This plane requires the use of the triangulation measurement principle. As the trajectory of the needle could not be visualized, as in the longitudinal plane, the puncture site must be calculated on the basis of the Pythagorean Theorem: using an angle of entry of 45°, the distance to the transductor will be approximately the same as the depth in which the center of the vein is found. The differentiation between the venal and arterial vessels was conducted through the compression technique, and if doubt arose, a Pulsed-Wave Doppler was utilized.

For the evaluation and selection of the venal vessel, we utilized the RAPEVA method [15] in order to insert the most adequate peripheral venous device in the most adequate vessel. As this method indicates, mapping was performed in systematized steps, analyzing internal structures, discarding the areas of risk, and localizing the most adequate place for catheter insertion, also ensuring the viability of the catheter path.

### 2.3. Equipment

As shown in Figure 2, the following equipment was utilized: the ultrasound machine from emergency services, a LOGIQ P5 750VA from General Electric Healthcare with an 11 mHz linear probe. Peripheral venous catheters model InsyteTM Autoguard^TM^ with gauges (G) ranging from 14G to 26G, with the following measurements: 26G 0.62 × 19 mm; 24G 0.7 × 19 mm; 22G 0.9 × 25 mm; 20G 1.1 × 30 mm; 18G 1.3 × 48 mm. Aside from the above, the following equipment for cannulation was used: compressor; non-sterile gloves; antiseptic solution: 70% alcohol, chlorhexidine alcohol 2%; gauze; three-way valve or extension; 10 mL syringe; 10 mL saline solution; dressing for setting the line; conducting gel if an ultrasound was performed; and extraction hood, fitting, and analytical extraction tubes, if needed.

### 2.4. Sample Calculation and Type of Sampling

The calculation of the sample size was performed following the guidelines established by Walters [16] for studies designed to compare the effectiveness of a new treatment with a standard. The level of significance was set at 95% with a power of 90%, thus obtaining a sampling size of 68 patients, with 34 patients per group, and where the total N was 72 patients. The type of sampling utilized was non-probabilistic and incidental. Given the nature of the work, a double-blind approach was not utilized.

### 2.5. Target Population and Study Population

To select the population (Figure 3), we utilized the following eligibility criteria: inclusion criteria: being older than 18 years old, having a prior history of at least a venous catheter in the last year to be able to compare the level of pain with the VAS between the previous catheters, and that the veins were not palpable/visible by the nurse and consenting to participate in the study. The exclusion criteria were as follows: patients located in the pediatrics, and resuscitation areas, as well as gynecology emergencies, or patients who did not full mental faculties.

### 2.6. Analysis Strategy

The main objective of the analysis was to compare the two techniques evaluated in each of the dependent variables defined previously. The demographic variables were utilized to describe the population under study and to find possible biases.

Comparisons of means (Student’s *t*-test or Wilcoxon test) or proportions (Fisher’s test) were performed according to the variable studied. For the continuous variables, their normality was checked with the Shapiro–Wilk test to decide if parametric (*t*-test) or non-parametric (Wilcoxon) tests would be utilized for the comparison of means. Moreover, the possible interactions between the dependent variables were assessed through an ANOVA.

In second place, the same variables were assessed considering a stratification of the sample as a function of the 6 risk groups. For this, an ANOVA was performed for each of the dependent variables.

The statistical analysis were performed with the R statistic program [17].

### 2.7. Ethical Aspects

This research study contemplates Law 15/1999, from 13 December, on Protection of Personal Data from article 18.4 of the Spanish Constitution, and Law 41/2002, from 14th November, on the autonomy of the patient and rights and obligations with regard to clinical information and documentation, as well as Law 1/2003, from 28 January, of the Generalitat, on Patient Rights and Information of the Valencian Community. The study was conducted according to the Good Clinical Practices found in ICH E6 guide for Good Clinical Practices, 1 May 1996, in accordance with the Declaration of Helsinki. The patient’s informed consent was obtained, as well as the approval of the Ethics Committee on Research from the hospital.

## 3. Results

### 3.1. Sociodemographic and Clinical Characteristics

A total of 72 patients were included in the study, 38 assigned to the ST group and 34 to the UST group. Table 1 describes the sociodemograhic and clinical characteristics of the population studied. It was observed that the only variable which showed a significant difference between the two techniques was BMI, which was greater in the group in which the ultrasound technique was utilized. None of the qualitative variables showed a difference in the proportion between the two groups. Both groups had the same characteristics, and therefore were comparable. Of total population studied (72 patients), 44 were women (61%) and 28 were men. The mean age of the population was 68.55 years old. The data obtained for temperature and blood pressure were found to be within the normal ranges for an adult patient, 36.45 °C and 122/68, respectively.

### 3.2. Variables of Interest

Table 2 describes the variables of interest, which were compared between the two groups. It was observed that the number of previous punctures and time significantly decreased through the use of the UST. With respect to the intensity of the pain, while no significant differences were found in pain perception in the previous cannulation between the two groups, the intensity of the pain in the present cannulation was significantly less with the UST. This improvement, defined as the difference between previous and present pain, was also significant.

Complications were not observed in only eight patients from the ST group, as compared to 25 patients from the UST group. It is important to point out that the number of complications was greater than the number of patients, given that the same patient could have suffered from multiple complications.

The catheter gauge also increased with the use of the UST. No cases were reported that utilized 24G and 26G gauges in the UST, considered pediatric, and only one case used 22G. No cases were reported for 14G or 16G in any group.

The percent of the rate of success was also analyzed according to the number of tries. It was observed that with the use of UST, a greater percentage of cannulations was achieved after the first try (76%), with no cases found where it was achieved after the third, fourth, or fifth tries. With the ST, it was achieved after the third try in most of the cases (39%). In this case, the success rate at the first attempt was 16%.

Lastly, with respect to the area of puncture, it varied significantly between the two groups, as the puncture was mostly performed on the inside of the elbow (97%) with the UST, while for the ST, the blood vessels found in the forearm (28%) or the hand (15%) were most frequently chosen.

Moreover, a multivariate analysis was performed to successfully predict the catheterization success, according to the catheterization technique and other co-variables (BMI, temperature, impossibility of puncture, cytostatic treatments, NLIV, time, perception of difficulty, and area of puncture). For this, a binary logistic regression was performed, with the results shown in Table 3.

### 3.3. Ultrasound Technique Group

An observational study was performed with data from only the UST group. Figure 4 shows the relationship between the depth and the complications. The data on the *y*-axis are shown in centimeters. It is observed that ecchymosis was found at a lower depth (its mean is lower than the rest). Extravasation was found in seven patients, and no complications were found in 25 patients. A pairwise Wilcoxon test was performed to verify if the differences observed visually were significant, but values of 0.11 and 0.22 were obtained when ecchymosis was compared with none and extravasation, respectively. Therefore, a significant relationship was found to not exist between depth and possible complications.

Figure 5 shows the existing relationship between the vessel diameter and the catheter gauge. The *y*-axis shows values expressed in centimeters. The greater the diameter, the greater the gauge. A Wilcoxon test was performed to verify the significance of this trend. This was only performed between the 18G and 20G gauges, as only one 22G case was found. The *p*-value obtained with Wilcoxon’s test was *p* = 0.007, which indicates the existence of a significant difference between the vessel diameters and the 18G and 20G gauges.

## 4. Discussion

The objective of the present study was to determine the benefits of the use of US with difficult venous access patients as compared to a standard technique. Similar studies have performed a comparative analysis between both techniques, and most showed improvements in cannulation with the use of US or did not find significant differences with respect to specific variables, but none found negative results on the use of US for experienced and non-experienced personnel [4,5,18,19,20]. Thus, of the reviews consulted, some concluded that there was an improvement in venous cannulation through the use of US [21,22], and the rest did not find significant differences with respect to the amount of time needed, and the number of punctures, although differences were found in the rest of the variables, such as rate of success, the patients’ perception of pain, or their degree of satisfaction [23,24,25], thus providing evidence on an improvement through the use of US.

In the present study, just as in the similar studies mentioned above, significant differences were not found between belonging to a risk group or the non-risk group. A significantly different result was only found in users of illegal IV drugs, with respect to the number of previous punctures, time, or intensity of pain. This could be explained by the effect of the sample size, given that only 6 of the 72 cases studied belonged to this group. It could also be explained by the great vascular damage found in these patients, which implied the need for multiple punctures, and therefore the increase in the cannulation time. Likewise, a significantly greater perception of pain intensity was found in the non-obese group. This could be attributed to the subjective nature of this variable. Diabetic patients were included in the present study, although this population was only studied in one article, which analyzed the use of US and utilized the variable insulin-dependent diabetes as the cause of difficulties for the puncture [5], although another study did introduce this variable as a risk factor [2], and a last article did not obtain significant results between this variable and patients with other difficulties [7].

With respect to age or sex, no significant differences were found between groups, and therefore these were comparable. No differences were found between temperature and blood pressure in any of the groups, despite hypotension and hypothermia being considered as vasoconstriction factors. This lack of differences could be that the data obtained were found to be within the normal range for an adult patient, and thus differences were not statistically significant. These results coincide with the article by Panebianco et al. [26], which concluded that the success of venous cannulation did not depend on the demographic or vital signs of the patient, but instead on the visualization and assessment of the vessel through the use of ultrasound. Likewise, no significant differences were found between the diagnostic tags when the patient was released, or in the priorities, flowcharts, and discriminators according to the Manchester Triage System, between the groups. The results obtained also coincided with those found in a study by Gonzalez Peredo [27] on the profile of the patients who were admitted to ES. The care times established by the patients classified as Priority 2 and 3, 57% of the total participants in the present study, were 10 and 60 min, respectively, and thus the improvement observed in the venous cannulation times in the UST group contributed to the improvement of care of the patients with difficult access, according to the quality indicators of the Emergency Services proposed by the Spanish Society of Emergencies (SEMES). It should be considered that Triage Commissions do not exist in most of the Spanish hospitals [23], which could condition the extrapolation of the data to the services at other centers. As concluded from the analysis, both groups behaved similarly, had the same characteristics, and were therefore comparable. The only variable which showed a significant difference was BMI, which was utilized as a stratification variable and was studied individually.

The mean number of punctures decreased significantly, and the percentage of success after the first try also increased. When analyzing the studies in which significant differences were not found with respect to the number of tries [23,24,25], we found that they did point to an improvement in the rest of the variables through the use of US; morever, none of them pointed to more than three tries until successful cannulation, which coincides with our data. The success after the first try through the use of the standard technique is associated with smaller gauge catheters, mostly with 22G catheters, which would not be adequate with time-dependent pathologies that require therapy with a great volume of fluids. Descriptive studies on the use of US [20] place the rate of greater success after the first and second try with 20G and 18G catheters, coinciding with the results obtained.

The measurement of time in the studies consulted varied considerably, making their comparison difficult; some fixed the start of measurement from the search for vessels [18,19,27], and others from time of venipuncture once the vessel had been selected [5,28]. In the present study, the start of measurement was established from the start of vessel search to the verification of success through the blood backflow after cannulation, with a mean of time obtained for the ST group of 618 s as compared to 126 through the use of UST. In a study conducted previously on the use of the peripheral vessels in the ES at the center where the present study took place [29], 416 s were established for cannulation using the ST, without discriminating between easy and difficult access. Therefore, the mean time is comparable to that obtained in the present results, which contributes to the decrease of possible biases due to the experience or skill of the operator. The times obtained in the study by Constantino et al. [30] were found to be 13 min (780 s) through the use of the UST, and 30 min (1800 s) with the ST, which indicates an improvement of 17 min, measured from the start of vessel search to blood backflow. Likewise, in a study that measured the time needed from skin puncture of non-experienced operators, a mean of 11 s was obtained [28]. Similar means were obtained in a nationwide study that also compared the use of both techniques by operators without prior experience, with a mean of 16 s for the UST [19]. In general, the studies in which the UST was compared with the ST [21,22], aside from those already cited, concluded that the ultrasound-guided cannulation implied a reduction in time as compared to the traditional (standard) method. Those that did not show significant differences with respect to time, or did not analyze it, showed better results through the use of US in other variables such as pain or operator satisfaction [13,20,23,24,25,30].

With respect to pain, the results obtained show that the ST did not lead to improvements between previous and present experience. On the contrary, in the cases group, an improvement of 1.44 points was obtained. Given that no significant differences were found in previous experience between the UST and ST groups, it can be affirmed that the two study populations were comparable, independently of the subject who had performed the technique. A nationwide study that described the use of US in venous cannulation concluded that the level of pain experienced by the patient, with the use of a Likert scale, was moderate, and increased in the cases of failed attempts [20]. Another study also measured the pain intensity using this scale but did not find significant differences between the standard and ultrasound-guided cannulation [31]. Moreover, it did not find differences with respect to the time and number of punctures, and thus it could be inferred that pain should not vary between one technique and the other. Given that the pain intensity and the degree of satisfaction of the patient are related, it should be underlined that most of the studies consulted showed improvements in the degree of satisfaction through the use of the UST in peripheral venous cannulation [20,27,31,32], as observed with the use of a Likert scale that ranged from 0 to 10 points. To avoid biases related with the patient’s treatment by the nurse, we created a questionnaire in which the patient evaluated not only aspects related with the puncture technique, but also with the relationship and empathy of the nurse. The questionnaire was composed of five categorical questions from 1 to 5 points, with 1 as “completely disagree” and 5 as “completely agree”. To ensure that the method was reliable, we conducted a pilot study on a group of nursing staff from ES, approximately 25% of the sample size, in order to assess the practical applicability of the questionnaire (reading difficulty, time used, etc.). The data obtained showed that the nurse did not treat the patients included in the cases group better than the rest.

With respect to the complications, the variables that were directly related with venipuncture were considered: extravasation, ecchymosis, and hematoma, given that the study did not include a posterior follow-up. The results obtained show that the use of US favored the decrease in the complications studied. The complications studied in other articles were diverse, both in the short and long term. Some studies only reported cases of arterial punctures, which could be compared to ecchymosis and hematoma and nerve punctures [21,22,33], with improvements also observed with the use of the US, or the decrease in the need to depend on a central venous access, and its associated complications [4,32,34]. 

With the ST, the most commonly used gauge was 22G, without any cases reported for 18G. These results could be due to deeper veins normally having a greater diameter, with the use of US facilitating their visualization, and that the vessels chosen for the ST tended to be more superficial and visible in patients who presented difficult access, and therefore the diameter of the vessel could be smaller. The most utilized gauges in the UST were 20G and 18G. The results obtained in the present study coincide with diverse articles that analyzed the gauge used in the UST [20,33,34], but could not be compared with those obtained through the use of ST, given that these were observational studies.

The most common puncture area was the inside of the elbow, or antecubital fossa in the case of the UST, and the forearm for the ST. Again, this was due to the depth of the vessels that can be found with the use the UST, which are non-palpable or non-visible with the ST in patients with a difficult access. No cases were found of cannulation with the UST in the brachial area, just as in other studies consulted, which mention this aspect [35,36,37], as the vessels found here tend to be deeper, and the catheters used in the present study were short in length (45 mm for the 14G catheters, and 19 mm for the 26G ones). The most commonly used catheter in the UST was 30mm long. The results coincide with diverse studies that concluded that the antecubital fossa was the first-choice location for the UST [23,27,29,33].

Given the loss of the catheter, either through extravasation or accidental extraction, and to discover if there could be a possible complication associated to the UST due to the depth of the vessels, we performed an analysis to determine if these two variables could be associated. We started with the premise that a longer portion of the length of the peripheral catheter inserted is found outside of blood vessel given its length, but finally, a significant relationship between them could not be determined given the low number of complications found. Three studies conducted a duration analysis, comparing short and long catheters, through the use of US [10,36,37].

Their conclusion was that the long catheters had a lower risk associated with their use, with a lower survival time in vessels located at a depth of 0.4 cm, and a greater one in vessels located at 1.2 cm. Therefore, increasing the length of the catheter, or choosing vessels that are not too deep, located in the antecubital fossa and forearm, as compared to the brachial area, would resolve this complication. The diameter of the catheter did not prevent its loss in any of the articles cited. Many studies [33,37,38] determined that the maximum distance suggested from skin to vein was <16 mm, while <12 mm could be considered adequate. Those results are in agreement with those obtained in the present study, providing support to the lack of significance, as none of the veins that were cannulated were found below 16 mm. The importance of the use of US comes from knowing the distance to the vessel to select the catheter whose length will result in that at least two-thirds of it are inside the vein, thus minimizing its loss and increasing survival [18,29]. Other articles that conducted a comparative study between UST and ST [39,40,41] pointed out that the survival of the catheter after three days was greater when the UST was used on the antecubital fossa with 18G catheters, in agreement with the results from the present study.

Lastly, only in the cases group, evidence was found of the relationship of the diameter of the cannulated vessel and the catheter gauge. The analysis was conducted between the 20G and 18G catheters, given that only one case was found in which the 22G catheter was used. It was concluded that as the diameter of the vessel increased, a larger catheter was used. The 18G catheters were utilized in vein vessels that were 0.6 cm in diameter, and the 20G were used with those that were 0.4 cm. It should be pointed out that in the study by Witting et al. [42], it was estimated that the rate of success of the cannulation with the UST increased when the vessels had a diameter equal to or greater than 0.4 cm. Therefore, there was a direct relationship between the picture viewed on the screen, its measurement, and the posterior selection of the adequate gauge of the catheter. Being able to assess the dimensions of the vessel, its height, width, and diameter, in real time eases the selection of the most appropriate catheter according to the measurements obtained. The articles found that alluded to this relationship [35] determined that the viewing of the vessel before puncture increased the gauge of the catheter. These articles stated that the prior visualization to assess the difficulty in vein access could predict the adequate gauge for a successful cannulation.

The main limitation of the study was the scarce bibliography published on the subject in the area of nursing. The experimental studies found in the bibliographic search were American, and the operators were normally doctors and not nurses. It should be taken into account that peripheral venous cannulation, either with an UST or ST, is considered explorer-dependent, meaning that the technical skill of the operator can produce biases, so that only one nurse collected the sample data. Moreover, all the participants who needed the intervention of a second or third operator were eliminated. The patients with more than five punctures were eliminated from the study as well, as the help from another health professional was solicited starting with the fifth try. Likewise, the use of the VAS could have also created biases, as it is measured subjectively and the pain threshold of every patient is different, and there could also be a memory bias when the patient was asked about previous experiences. Moreover, a possible bias was the slightly higher BMI in the cases group. Presently, the use of the ultrasound-guided technique is not an established routine with patients with difficult access, and therefore there is only a small number of nurses who are trained on the subject, and it is also not part of the academic curriculum in the Nursing degree. With respect to the classification of risk groups, during the period of data collection, no studies existed on validated guides to predict or identify the profile of the patient with difficult venous access, such as the study by Van Loon et al., which will be considered for future studies [43]. Given the nature of the study, it was impossible to conduct it as a double-blind study, and posterior follow-up was not performed; therefore, the complications were not assessed in the short or long term. The non-probabilistic sampling was also a limitation of the study.

## 5. Conclusions

The use of the ultrasound decreased the number of punctures, the time, and the complications derived from venipuncture; the number of punctures before success; and the pain intensity of the patient with respect to the standard technique. Likewise, the success rate and the gauge of the catheter increased.

In the group with which the UST was utilized, we did not find a relationship between the complications and the depth of the vessels, although a relationship was determined between the selection of the catheter gauge and the visualization and measurement of the diameter of the vein structure through the use of the ultrasound, obtaining venous access according to the dimensions of the vessel.

Therefore, and on the basis of the results obtained, the use of ultrasound is recommended for the cannulation of peripheral venous devices in patients with difficult venous access, given the benefits in clinical practice and the advantages it presents as compared to the standard technique in the measured variables.

## Figures and Tables

**Figure 1 healthcare-10-00261-f001:**
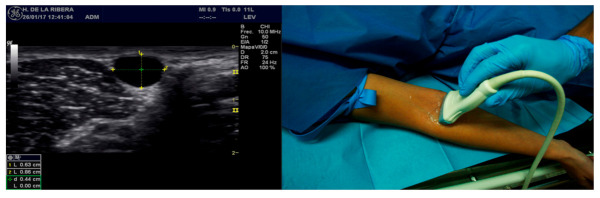
Measurement of the vein through a transverse plane. Source: author created.

**Figure 2 healthcare-10-00261-f002:**
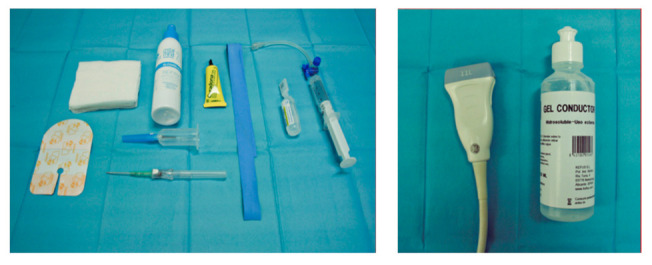
Equipment needed for ultrasound-guided peripheral venous cannulation. Source: created by author.

**Figure 3 healthcare-10-00261-f003:**
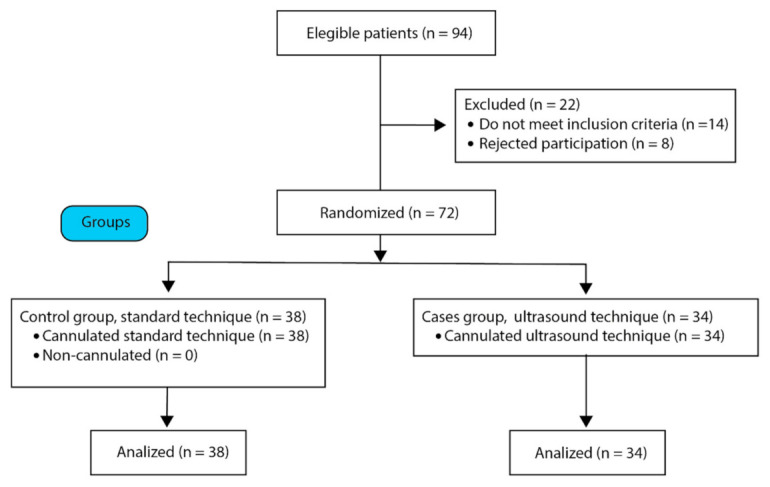
Flow diagram of the strategy used for the selection of the study population. Source: created by authors.

**Figure 4 healthcare-10-00261-f004:**
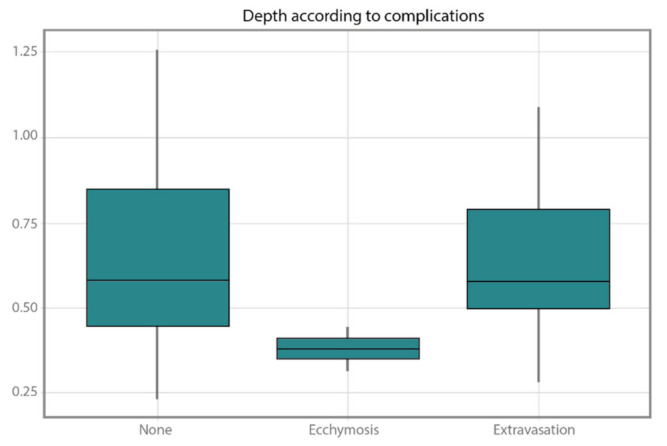
Relationship between the depth of the vessel and complications shown after ultrasound-guided cannulation.

**Figure 5 healthcare-10-00261-f005:**
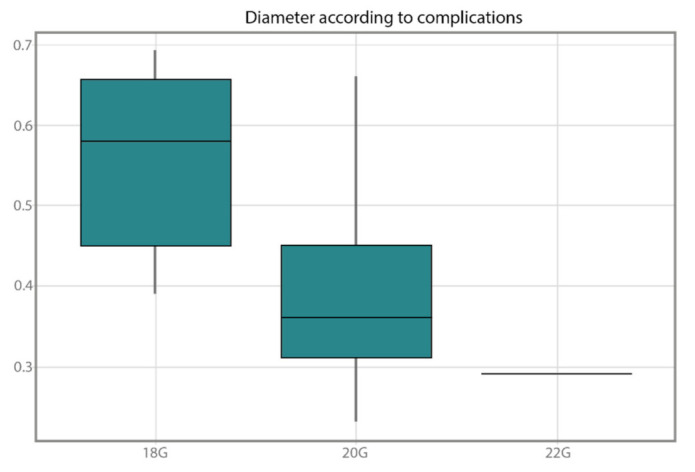
Relationship between vessel diameter and catheter gauge utilized in the ultrasound-guided cannulation.

**Table 1 healthcare-10-00261-t001:** Sociodemographic characteristics and risk groups of the study population.

Variable		Total (*n* = 72)	Controls (*n* = 38)	Cases (*n* = 34)	*p*-Value
Sex *	Man	28 (39)	16 (47)	12 (35)	0.6318
	Woman	44 (61)	22 (53)	22 (65)	
Age ^+^		68.55 ± 17.56	68.89 ± 18.77	68.18 ± 16.67	0.5384
BMI ^+^		29.49 ± 6.18	27.51 ± 5.56	31.52 ± 6.22	0.0058
Temperature ^+^		36.45 ± 0.74	36.52 ± 0.79	36.38 ± 0.67	0.7995
BP ^+^	Systolic	122.19 ± 26.24	124.39 ± 24.37	119.74 ± 28.35	0.4598
	Diastolic	68.71 ± 16.99	68.47 ± 13.75	68.97 ± 20.22	0.6557
Triage priority *	2	13 (18)	8 (21)	5 (15)	0.5782
	3	28 (39)	16 (42)	12 (35)	
	4	31 (43)	14 (37)	17 (50)	
Obesity *		31 (43)	12 (32)	19 (56)	0.028
DM *		26 (36)	9 (24)	15 (44)	0.028
Cytostatic treatment *	Never	52 (72)	28 (74)	24 (71)	
	Not currently	7 (10)	5 (13)	2 (6)	
	Yes currently	13 (18)	5 (13)	9 (23)	0.355
Anticoagulant treatment *		36 (50)	17 (45)	19 (56)	0.479
NLIV Drug user *		6 (8)	5 (13)	1 (3)	0.203
Inability Puncture *	None	51 (71)	27 (71)	24 (71)	0.965
	CVA	8 (11)	3 (8)	5 (15)	
	Ganglion extirpation	5 (7)	3 (8)	2 (6)	
	AVF	4 (6)	2 (5)	2 (6)	
	Skin lesions	4 (6)	3 (5)	1 (3)	

^+^: mean ± SD; *: *n* (%); BMI: body mass index; BP: blood pressure; DM: diabetes mellitus; NLIV: non-legal intravenous; CVA: cerebrovascular accident; AVF: arteriovenous fistula.

**Table 2 healthcare-10-00261-t002:** Comparison of the variables between the groups: controls (standard technique) and cases (ultrasound technique).

Variable		Total (*n* = 72)	Controls (*n* = 38)	Cases (*n* = 34)	*p*-Value
Number of previous punctions ^+^		2.12 ± 1.24	2.92 ± 1.19	1.23 ± 0.43	<0.001
Time (seconds) ^+^		385.93 ± 79.58	618.34 ± 387.16	126.17 ± 101.09	<0.001
Pain ^+^	Previous	6.36 ± 1.87	6.65 ± 1.65	6.02 ± 2.07	0.157
	Present	5.62 ± 2.09	6.55 ± 1.70	4.58 ± 2.03	0.023
	Improvement	0.74 ± 1.2	0.11 ± 0.39	1.44 ± 2.12	<0.001
Complications *	None	33	8	25	<0.001
	Ecchymosis	28	26	2	
	Hematoma	11	11	0	
	Extravasation	19	12	7	
Prior punctures *	1	32 (44)	6 (16)	26 (76)	<0.001
	2	14 (19)	6 (16)	8 (24)	
	3	15 (21)	15 (39)	0 (0)	
	4	7 (1)	7 (18)	0 (0)	
	5	4 (0.6)	4 (11)	0 (0)	
Catheter gauge *	18G	14 (19)	0 (0)	14 (41)	<0.001
	20G	26 (36)	7 (18)	19 (56)	
	22G	20 (28)	19 (50)	1 (3)	
	24G	10 (14)	10 (26)	0 (0)	
	26G	2 (3)	2 (5)	0 (0)	

^+^: mean ± SD; *: *n* (%).

**Table 3 healthcare-10-00261-t003:** Multivariate analysis to predict catheterization success.

Equation Variables						
	B	Standard Error	Wald	gl	Sig.	Exp(B)
Technique	56.674	3356.953	0.000	1	0.987	4102609739702777600000000.000
BMI	1.505	475.736	0.000	1	0.997	4.505
Temperature	30.429	2145.174	0.000	1	0.989	16418261462635.236
Impossibility of puncture	46.219	2367.183	0.000	1	0.984	118195291007880700000.000
Cytostatic drugs	14.084	4910.388	0.000	1	0.998	1307353.215
DM	−0.668	13,073.356	0.000	1	1.000	0.513
NLIV	−32.661	22,003.851	0.000	1	0.999	0.000
Time	−0.663	33.719	0.000	1	0.984	0.515
Difficulty	−9.505	1078.693	0.000	1	0.993	0.000
Area of puncture	−28.355	8800.896	0.000	1	0.997	0.000
Constant	−1024.217	71,753.598	0.000	1	0.989	0.000

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
