# Peer review of "Use of the Ultrasound Technique as Compared to the Standard Technique for the Improvement of Venous Cannulation in Patients with Difficult Access"

_healthcare, 2022, doi:10.3390/healthcare10020261_

Round 1
Reviewer 1 Report
Congratulations to the authors with this manuscript. This study focused on an interesting and relevant problem in modern healthcare, the difficulty to obtain peripheral intravenous access. Not only for patients admitted to the Emergency Department, but for the entire hospitalized population. To further improve the manuscript, and to consider it for publication, I suggest some corrections and additions to it.
The first chapter gives a clear introduction of the subject studied in this manuscript and an outstanding clarification of the problems with difficult intravenous cannulation. Many risk factors for difficult intravenous cannulation are identified in previous studies, many research was performed on this. There are some published studies that merged these risk factors and created prediction models to calculate a patients individual risk profile for failed cannulation on the first attempt of the presence of a difficult intravenous access. The authors mentioned in the manuscript “these patients are defined as difficult-to-access or difficult venous access patients and are a placed in risk groups according to different studies” (page 2, lines 53 – 54). This information is not complete and may be incorrect regarding the used references. In my opinion, the most important and best useful tool to predict a patients’ difficulty for obtaining intravenous access is the A-DIVA scale, as published by Van Loon et al. (2019, Journal of Clinical Medicine). Furthermore, this scale can by a useful tool in this research project, to classify patients and guide the application of ultrasound.
To continue, the aim of the study fits with the information presented in the introduction section (page 2, lines 85 – 90). Though, to make a bridge to the methods section and to draw attention to the study design, the authors can consider to mention that this was a randomized study.
An experimental, randomized and cross-sectional study was chosen. This design can positively influence the level of evidence of the study. Nonetheless, there is a major risk for (selection) bias in the current study, which is introduced by the way subject are allocated to either intervention or control group by using a non-probabilistic randomization strategy (page 2, lines 93 – 95). How did the authors handle the risk for (selection) bias in the current study?
To add on this, patients with difficult venous access were defined as those whose veins were not visible and/ or palpable, as determined by nurses (page 3, lines 100 – 102). These criteria are quite subjective, especially in the Emergency Department. I can imagine that many other variables (body temperature, hemodynamic condition, pain) are common in the Emergency Department population, affecting venous size. In my opinion, using visibility and palpability as the only factors to label a patient as difficult to cannulate does not create a reliable reflection of a patients’ real risk for a difficult intravenous access. Simply, the objective of this study was to determine the benefits of the use of ultrasound in patients with a difficult intravenous access, so an unbiased classification of a patients should be used.
For intravenous cannulation, 14 to 26 gauged catheters are used (page 3, lines 135 – 136). Can the authors provide information about the length of the catheters, because this is affection cannulation success. This is particularly relevant in patients in whom an ultrasound-guided technique was applied, in which deeper situated veins can be identified and cannulated, but longer sized catheters are needed. In fact, using needles that are too small in length (or cannulating too deep veins) can achieved more cases of failed cannulation based on the use of the wrong equipment.
Please provide information about the approval of the Ethics Committee on Research from the hospital in terms of a reference number and date (page 5, lines 185 – 187).
Please give p-values for all factors in Table 1.
The most crucial information is missing, for me. The authors write that the number of previous punctures is significantly lower after applying the ultrasound-guided technique, when compared to the traditional landmark approach, as shown in Table 2 (page 6, lines 207 – 208). Apart from this important observation, it appeals more to the imagination of the reader to see the first attempt success rate (number of successful cannulation, divided by the total number of cannulation, and multiplied by 100%). I suggest to add this to the results section.
I would like to advice the authors to control for the included and measured variables and do a multivariate analysis with the outcome of interest (cannulation success) and the other factors in the analysis. Without that analysis, readers cannot judge whether or not the use of ultrasound truly increases first attempt cannulation success. I expect there is enough data to run a multivariate analysis, and the authors will have no problem finding a competent statistician to help them with this. The same applies to complications regarding the applied strategy, authors must explore if a true effect of ultrasound can be demonstrated. I can imagine that the risk for complications increases after the insertion of a larger sized catheter, which is more often inserted with the use of ultrasound. There only was a Wilcoxon Signed signed rank test used to look for differences (which are denoted as relationships by the authors), although regression analysis was not performed.
Finally, I miss an advice from the authors based on the study results in the manuscript in the conclusion section (page 12, lines 443 – 452). Should ultrasound be used? And to what patients should ultrasound be applied?
Author Response
First of all, we would like to thank you for taking the time to send us your suggestions, which will undoubtedly help us to improve the manuscript. In the file attached below, we offer the modifications made according to your indications, and we are at your disposal for any future revisions you may consider appropriate.
Kind regards,

Reviewer 2 Report
Dear Editor,
Thank you for asking me to review the paper by Angeles Rodrigues-Herrera et al.
In my opinion the manuscript should be extensively revised. Please find my points below :
The quality of manuscript language is very poor, especially in Discussion section making it very difficult to understand.
Materials and method :
There no description of limbs examination and cannulation technique.
Were all staff members equally expired ? What was the judgement criteria ?
What was the reason of trauma or gynaecology patients exclusion ?
Is there a mistake in UST limb in Figure 1 ? Is the number of patients of 34 and cannulated vessels 38 ?
Patients should be analysed according also parameters including hears rate, respiratory rate, GCS etc.
Patent AVF arm in haemodialyzed patient must not be used for cannulation other that for dialysis purposes ! I believe there were limbs with non-functioning AVF used for veins cannulation.
Results :
The cannulated vessel site must be included into analysis. It differs significantly !
Why UST patients have had cannulated elbow or antecubital vessels mostly ? Was it caused by to extensive compression with the probe on the forearm and hands ?
Discussion : This section must be edited by native speaker as it is barely understandable !
Author Response

(The authors gave the same response as above.)

Round 2
Reviewer 1 Report
Congratulations with this revised version of the manuscript. I see an optimized and improved version of it, that is ready for publication. I have one suggestion and important remark: the authors must check and correct the information in Table 3 (multivariate analysis to predict catheterization success). I suppose there was a mistake made in the presentation of the Exp(B) for some factors (technique, temperature, impossibility of puncture and cytostatic drug). The presented Exp(B) are too large, especially when compared to the presented B.
Author Response
We appreciate your contributions, as they help us to improve the version of our manuscript. Below is the response to your request and we hope it complies with your requirements.
Thank you very much,
Kind regards,

Reviewer 2 Report
Dear Editor,
I have read the revised version of the manuscript. Authors have edited the paper extensively, improved statistic failures as well as improved the Discussion section. In my opinion the present form is acceptable for publication.
Author Response
We appreciate your input, as it has helped us to improve the version of our manuscript.
Thank you very much,
Kind regards,